

# The role of the underlying event in the charm-baryon enhancement observed in pp collisions at LHC energies

Zoltán Varga[1,2⋆], Anett Misák[1,3] and Róbert Vértesi[1]

**1** Wigner RCP, Budapest, Hungary
**2** Budapest University of Technology and Economics, Budapest, Hungary
**3** Eötvös University, Budapest, Hungary

⋆ varga.zoltan@wigner.hu

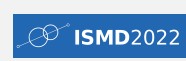

## Abstract

We study the enhanced production of $\Lambda_c$ charmed baryons relative to that of charmed $D^0$ mesons in proton–proton collisions at LHC energies. We simulated collision events with the enhanced color-reconnection model in PYTHIA 8 MC generator and propose measurements based on the comparative use of different event-activity classifiers to identify the source of the charmed-baryon enhancement. We demonstrate that in this enhanced color-reconnection scenario the excess production is primarily linked to the underlying event and not to the production of high-momentum jets.

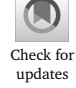

## 1 Introduction

The factorization hypothesis states that the production cross-section of heavy-flavor hadrons can be calculated as the convolution of three independent terms: the parton distribution function of the colliding hadrons, the production cross sections of the heavy-quarks in the hard partonic process, and finally the fragmentation functions of the heavy-flavor quarks into the given heavy-flavor hadron species. The fragmentation function has been traditionally treated as universal, i.e. independent of the collision system.

Recent charmed-baryon measurements show a low-momentum enhancement over model predictions which are based on $e^+e^-$ collisions, which challenges this traditional assumption of universality [1, 2]. One of the latest measurements also shows that the charm-baryon enhancement depends on the final-state multiplicity of the collision event [3]. Several scenarios have been proposed to explain the emerging pattern, including string formation beyond leading color, the so-called enhanced color re-connection (CR) [4], which provides a qualitatively correct description of these findings for pp collisions.

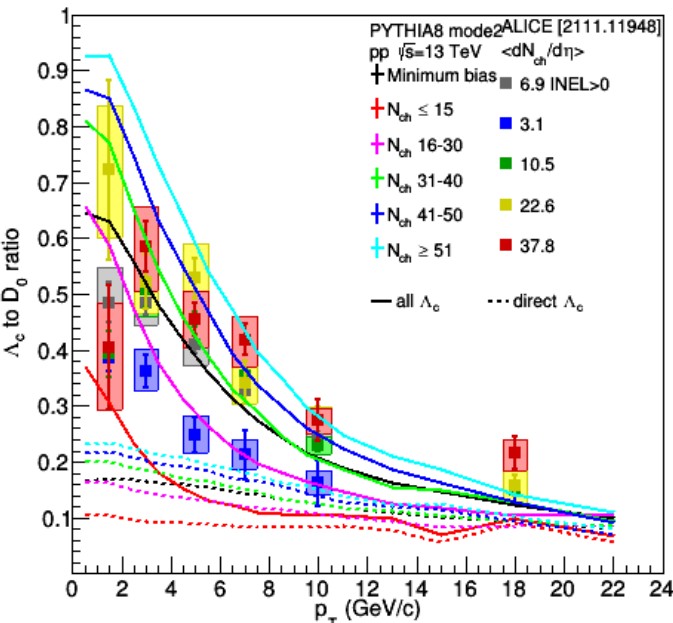

Figure 1: The $\Lambda_c$ to $D^0$ ratios in function of $p_T$ for different event multiplicity bins. The ratios for only the directly produced $\Lambda_c$ are also shown with a dotted line.

In this proceedings, we briefly recapitulate our results on investigating the charm-baryon enhancement with PYTHIA 8 Monte-Carlo generator and enhanced color-reconnection models [5] and compare the calculations to new results from ALICE [3]. We used the enhanced CR mode 2 as it is found to describe data the best [2]. Note that PYTHIA with the built-in CR that does not feature color junctions and does not predict any enhancement.

We propose measurements based on several event-activity classifiers to identify the source of the charm-baryon enhancement.

## 2 Results

We focus on two event classifiers in this proceedings, both of them are used when the highest transverse-momentum (trigger) particle in the acceptance has $p_T^{\text{trig}} > 5$ GeV/$c$ in the selected event. One of them is the transverse event activity classifier $R_T \equiv N_{\text{ch}}^{\text{trans}}/\langle N_{\text{ch}}^{\text{trans}}\rangle$ [6], where $N_{\text{ch}}^{\text{trans}}$ is the charged-hadron multiplicity in the transverse region. It is defined with the azimuth angle relative to the trigger hadron being $\frac{\pi}{3} < |\Delta\phi| < \frac{2\pi}{3}$ within $|\eta| < 1$.

The other one is the near-side cone activity classifier $R_{\text{NC}} \equiv N_{\text{ch}}^{\text{near-side cone}}/\langle N_{\text{ch}}^{\text{near-side cone}}\rangle$ [5], where $N_{\text{ch}}^{\text{near-side cone}}$ is the charged-hadron multiplicity in a narrow cone around the trigger particle, where $\sqrt{\Delta\phi^2 + \Delta\eta^2} < 0.5$. Due to the constraints of space in this proceedings, we refer to the description of the remaining details of the analysis method in [5].

In Fig. 1 we show the $\Lambda_c$ to $D^0$ ratios for different charged event multiplicity bins using CR Mode 2 in PYTHIA 8. The minimum-bias results, i.e. where no event-activity selection was applied, are compared to ALICE measurements [3] for reference. We note that the trends are different for the $\Lambda_c$ produced directly and those coming from $\Sigma_c$ and this needs further experimental investigation.

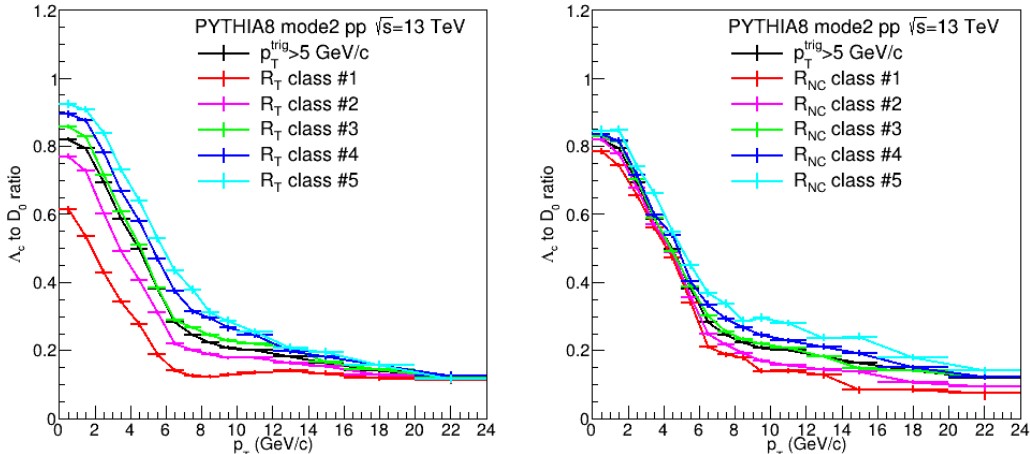

Figure 2: The $\Lambda_c$ to $D^0$ ratios are plotted in function of $p_T$ for different $R_T$ bins (left) and $R_{NC}$ bins (right).

In Fig. 2 we plot the $\Lambda_c$ to $D^0$ ratios in function of $p_T$ for the $R_T$ and $R_{NC}$ event activity classifiers. We observe that the $\Lambda_c$ enhancement significantly depends on the underlying event activity. However, in the coalescence regime $2 < p_T < 8$ GeV/$c$ there is no difference among the near-side cone activity classes, which measure the jet region activity. (Note that an ordering with event activity is present in the higher transverse-momentum range which can be a bias as the trigger is often provided by the heavy-flavor process.) We can draw the conclusion that the mechanism behind the enhancement in the coalescence regime is connected to the underlying event, rather than the jet region. The exact values of the activity classifier bins can be found in [5].

## 3 Conclusion

We simulated the $\Lambda_c$ to $D^0$ yield ratios with PYTHIA 8 for the enhanced color re-connection model. Our results were compared to ALICE experimental data for different event multiplicity classes. A difference can be observed between the $\Lambda_c$ yields coming from $\Sigma_c$ and those produced directly, which needs to be investigated further experimentally.

By investigating the enhancement with different event activity classifiers, we conclude that within the enhanced color re-connection model, the excess $\Lambda_c$ production is connected to the underlying event and not the jet production [5]. These observables provide a unique opportunity in the upcoming measurements from the high-luminosity LHC Run3 phase to understand charm fragmentation mechanisms, and will serve as valuable means for further model development.

## Acknowledgements

**Funding information** This work was supported by the Hungarian National Research, Development and Innovation Office (NKFIH) under the contract numbers OTKA FK131979 and K135515, and the NKFIH grant 2021-4.1.2-NEMZ_KI-2022-00007. The authors acknowledge the computational resources provided by the Wigner Scientific Computing Laboratory (WS-CLAB) and research infrastructure provided by the Eötvös Loránd Research Network (ELKH).

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
