# Peer review of "The role of the underlying event in the charm-baryon enhancement observed in pp collisions at LHC energies"

_SciPost Physics Proceedings, doi:SciPost Phys. Proc. 15, 014 (2024)_

## Round 1 · Referee Report · Anonymous · 2022-11-28

Strengths
1. A very accessible introduction, making the topic clear to all readers
2. A reasonably clear presentation of the link between activity classes and
Weaknesses
1. The role or distinction between Pythia models is not made clear: is this phenomenon also seen with other Pythia CR models? Is this model realistic compared to data?
2. The connection between the ratio plots with different activity classes and the conclusions drawn seem unclear: enhancements seem to happen in different pT ranges for both the different activity classifiers, with the indirect-production dominance falling into one of those ranges. More explanation and discussion of these interesting features would be valuable.
Report
The presentation is generally good, but more questions are introduced than answered in the results section. Expanding upon this to better explain the conclusions drawn from the plots would make it a much stronger contribution. (The page limit is relaxed for corrections after review.)
Requested changes
1. The definition of the trigger particle is not quite clear: what if more than one particle has pT > 5 GeV?
2. constrains -> constraints
3. p2: The meaning of CR Mode 2 is not given: this is completely obscure to any non-expert in Pythia CR modes.
4. Trapped text below Fig 1; would be clearer with Fig 1 positioned at the [b]ottom rather than [h]ere.
5. p3: the "jet region activity classes" have not been mentioned... previously this variable was the near-side cone activity" classifier. Either use the same name in both places, or introduce the "jet" nomenclature as a synonym at the definition point.
6. p3: the Lambda_c enhancement (i.e. ratio increase with event class) does actually appear for both the transverse and near-side classifier binnings, it just happens at low pT for the transverse classes, and at higher pT for the (jet-like) near-side. The connection to the bias of near-side particles to themselves be higher-pT seems clear, but it is maybe also important that the dividing line between these pT regimes appears to be the region dominated by Sigma_c production in min bias? These aspects should be discussed to some extent, as otherwise the images seem to contradict the text, or at least suggest possible alternative or more subtle explanations.
Author: Zoltan Varga on 2022-12-02 [id 3100]
(in reply to Report 1 on 2022-11-28)We thank the Referee for the positive overall evaluation of our manuscript, and for the valuable suggestions. We addressed all the comments and questions by the Referee. We're resubmitting the revised manuscript with the requested changes implemented.
We agree that the results in our contribution raise a few questions. We find remark 6 especially useful and we extended the discussion on that.
We are also carrying out further studies that do not fit into these proceedings. Some of the questions will eventually be answered by high-quality data from the LHC Run3 period.

---

## Round 2 · Author Response

We thank the Referee for the positive overall evaluation of our manuscript, and for the valuable suggestions. We addressed all the comments and questions by the Referee. We're resubmitting the revised manuscript with the requested changes implemented.
We agree that the results in our contribution raise a few questions. We find remark 6 especially useful and we extended the discussion on that.
We are also carrying out further studies that do not fit into these proceedings. Some of the questions will eventually be answered by high-quality data from the LHC Run3 period.
We agree that the results in our contribution raise a few questions. We find remark 6 especially useful and we extended the discussion on that.
We are also carrying out further studies that do not fit into these proceedings. Some of the questions will eventually be answered by high-quality data from the LHC Run3 period.

---

## Round 2 · List of Changes

We addressed all the comments of the Referee as follows:
1 Introduction
We added the sentences "We used the enhanced CR mode 2 as it is found to describe data the best [2]. Note that PYTHIA with the built-in CR that does not feature color junctions and does not predict any enhancement." before the last sentence.
2 Results
We changed the 2nd sentence as " the highest transverse-momentum (trigger) particle in the acceptance has pT trig > 5 GeV/c in the selected event. "
In the last paragraph we changed the 2nd sentence, added a 3rd and changed the 4th sentence as:
"However, in the coalescence regime 2 < pT < 8 GeV/c there is no difference among the near-side cone activity classes, which measure the jet region activity. (Note that an ordering with event activity is present in the higher transverse-momentum range which can be a bias as the trigger is often provided by the heavy-flavor process.) We can draw the conclusion that the mechanism beyond the enhancement in the coalescence regime is connected to the underlying event, rather than the jet region."
We fixed a typo "constraint".
We eliminated the trapped text by moving the Figure 1 to the end of the document.
1 Introduction
We added the sentences "We used the enhanced CR mode 2 as it is found to describe data the best [2]. Note that PYTHIA with the built-in CR that does not feature color junctions and does not predict any enhancement." before the last sentence.
2 Results
We changed the 2nd sentence as " the highest transverse-momentum (trigger) particle in the acceptance has pT trig > 5 GeV/c in the selected event. "
In the last paragraph we changed the 2nd sentence, added a 3rd and changed the 4th sentence as:
"However, in the coalescence regime 2 < pT < 8 GeV/c there is no difference among the near-side cone activity classes, which measure the jet region activity. (Note that an ordering with event activity is present in the higher transverse-momentum range which can be a bias as the trigger is often provided by the heavy-flavor process.) We can draw the conclusion that the mechanism beyond the enhancement in the coalescence regime is connected to the underlying event, rather than the jet region."
We fixed a typo "constraint".
We eliminated the trapped text by moving the Figure 1 to the end of the document.

---

## Editorial Decision

published